# *Moringa oleifera* Leaf Extract Upregulates Nrf2/HO-1 Expression and Ameliorates Redox Status in C2C12 Skeletal Muscle Cells

**DOI:** 10.3390/molecules26165041

**Published:** 2021-08-20

**Authors:** Guglielmo Duranti, Mariateresa Maldini, Domenico Crognale, Katy Horner, Ivan Dimauro, Stefania Sabatini, Roberta Ceci

**Affiliations:** 1Laboratory of Biochemistry and Molecular Biology, Department of Movement, Human and Health Sciences, Università degli Studi di Roma “Foro Italico”, Piazza Lauro de Bosis 6, 00135 Rome, Italy; stefania.sabatini@uniroma4.it (S.S.); roberta.ceci@uniroma4.it (R.C.); 2SCIEX Italia s.r.l., Via Montenapoleone, 8, 20121 Milano, Italy; mariateresa.maldini@sciex.com; 3Institute for Sport & Health, School of Public Health, Physiotherapy and Sports Science, University College Dublin, D04 V1W8 Dublin, Ireland; domenico.crognale@ucd.ie (D.C.); katy.horner@ucd.ie (K.H.); 4Laboratory of Biology and Human Genetic, Department of Movement, Human and Health Sciences, Università degli Studi di Roma “Foro Italico”, Piazza Lauro de Bosis 6, 00135 Rome, Italy; ivan.dimauro@uniroma4.it

**Keywords:** *Moringa oleifera* leaf extract (MOLE), C2C12 skeletal muscle cells, redox status, enzymatic antioxidant system, oxidative metabolism

## Abstract

*Moringa oleifera* is a multi-purpose herbal plant with numerous health benefits. In skeletal muscle cells, *Moringa oleifera* leaf extract (MOLE) acts by increasing the oxidative metabolism through the SIRT1-PPARα pathway. SIRT1, besides being a critical energy sensor, is involved in the activation related to redox homeostasis of transcription factors such as the nuclear factor erythroid 2-related factor (Nrf2). The aim of the present study was to evaluate in vitro the capacity of MOLE to influence the redox status in C2C12 myotubes through the modulation of the total antioxidant capacity (TAC), glutathione levels, Nrf2 and its target gene heme oxygenase-1 (HO-1) expression, as well as enzyme activities of superoxide dismutase (SOD), catalase (CAT), glutathione peroxidase (GPx) and transferase (GST). Moreover, the impact of MOLE supplementation on lipid peroxidation and oxidative damage (i.e., TBARS and protein carbonyls) was evaluated. Our results highlight for the first time that MOLE increased not only Nrf2 and HO-1 protein levels in a dose-dependent manner, but also improved glutathione redox homeostasis and the enzyme activities of CAT, SOD, GPx and GST. Therefore, it is intriguing to speculate that MOLE supplementation could represent a valuable nutrition for the health of skeletal muscles.

## 1. Introduction

*Moringa oleifera* Lam., a member of the tropical family of Moringaceae, is commonly referred to as a “miracle tree” for its numerous nutritive, medicinal and industrial potentials.

The bioactive compounds found in *Moringa oleifera* leaves, which are the parts of the plant more frequently utilized, include tannins, saponins, flavonoids, terpenoids and glycosides. Many of these molecules have been shown to be beneficial as antioxidants, antimicrobial, anti-carcinogenic agents [1,2], as well as to be effective in treating several chronic pre-pathological conditions such as hypercholesterolemia, insulin resistance and inflammation, of which the onset is based on the increase in reactive oxygen species (ROS) [3,4,5,6].

Particularly, phenolic acids present at high concentrations in extracts of freeze-dried leaves of *Moringa oleifera* are known to act as primary antioxidants either inactivating lipid-free radicals or preventing the decomposition of hydroperoxides into free radicals [7,8,9].

The antioxidant properties of *Moringa oleifera* have been extensively investigated in different systems [10,11,12,13]; however, their effects on redox homeostasis of muscle cells have not been investigated. This topic is particularly interesting because skeletal muscle is a tissue where ROS are continuously produced by several sources (i.e., mitochondria, NADPH oxidases, phospholipase-A2, xanthine oxidase), especially during contractile activity.

Moreover, skeletal muscle is considered the largest insulin-sensitive tissue in the body, and characterized by sudden metabolic demands leading to an increase in mitochondrial O_2_ consumption to match the increase in oxidative metabolism. These changes in substrates oxidation could shift the cellular redox environment to a more oxidized state, potentially causing damage to macromolecules [14,15].

To date, it has been widely accepted that skeletal muscle contraction leads to a consistent increase in free radical content that, if not adequately contrasted, might induce a decreased mitochondrial respiratory control, loss of sarcoplasmic and endoplasmic reticulum integrity, and increased levels of lipid peroxidation products [16,17]. Furthermore, ROS production is deregulated during muscle aging and pathological processes [18,19]. It is noteworthy that the excessive ROS levels observed during aging are also involved in triggering sarcopenia [20]. On the other hand, healthy skeletal muscle is able to counteract exercise or metabolic oxidative stress by inducing the antioxidant enzymatic system.

Our previous study showed that *Moringa oleifera* leaf extract (MOLE), due to the presence of glucosinolates, flavonoids and phenolic acids, improved oxidative capacity in C2C12 skeletal muscle cells, as displayed by the increased reliance on fat metabolism through the activation of the SIRT1-PPARα pathway [21]. Indeed, a number of studies has shown that SIRT1, besides being a critical energy sensor able to modulate transcription factors involved in muscle mitochondrial biogenesis and function [22,23], is also involved in maintaining cellular redox homeostasis [24,25]. In fact, this protein is very effective in attenuating oxidative stress through the deacetylation of transcription factors, including nuclear factor erythroid 2-related factor (Nrf2) [26], and in such a way potentiates the cellular defense system. Nrf2, upon a redox status imbalance, dissociates from the sequestration complex and translocates to the nucleus where it interacts with the antioxidant-responsive element (ARE) of antioxidant genes. These events lead to the transcriptional activation of its target genes such as heme oxygenase-1, superoxide dismutase, glutathione peroxidase and glutathione S-transferase [27,28,29].

Based on these findings, we hypothesized that MOLE could allow skeletal muscle cells to improve their redox status counteracting efficiently ROS induced by oxidative metabolism through the antioxidant enzymatic system induction.

To our knowledge, there is no information about the influence of MOLE on redox balance in skeletal muscle cells. For the first time, here, we proposed an in vitro experimental design without the challenge of an oxidative insult in order to evaluate the effects of MOLE per se.

In particular, differentiated C2C12 skeletal muscle cells were treated with MOLE and analyzed for: (a) the antioxidant regulatory Nrf2 and HO-1 protein levels, the total antioxidant capacity (TAC), an assay that measures lipo- and hydro-philic antioxidants; (b) GSH homeostasis as markers of redox status; (c) enzyme activities of superoxide dismutase (SOD), catalase (CAT), glutathione peroxidase (GPx) and glutathione transferase (GST) engaged in antioxidant enzymatic defense; (d) lipid peroxidation (TBARS) and protein carbonyls (PrCar) as markers of oxidative damage.

## 2. Materials and Methods

All chemical reagents, unless otherwise specified, were purchased from Sigma-Aldrich Chemical (St. Louis, MO, USA).

### 2.1. Methanolic Extract of Moringa oleifera Leaves

One gram of *Moringa oleifera* leaf powder (PureBodhi Nutraceuticals Ltd., Cambridge, UK) was sonicated (Vibra-Cell CV 18 SONICS VX 11, Sonics & Materials, Newtown, CT, USA) in 10 mL of methanol 100% twice for 10 min at +4 °C. The extract was then centrifuged (2000× *g* for 10 min at +4 °C), collected and stored at −20 °C (stock solution).

### 2.2. Qualitative Profiling of MOLE Extract

Qualitative profiling of tested MOLE extract was obtained by using a SCIEX X500B QTOF mass spectrometer (UHPLC-QTOF) (AB SCIEX GmbH, Landwehrstraße 54, Darmstadt, Germany). LC and MS conditions were utilized as previously described [21].

Briefly, 10 μL of diluted extract (1:10; *v*/*v* with buffer A) was tested. Separation was performed using a Phenomenex Luna Omega Polar C18 (150 × 4.6 mm, 3 μm, Phenomenex Inc. Via M. Serenari, 15/D, Castel Maggiore (BO), Italy) analytical column (maintained at 40 °C). An amount of 0.1% formic acid in water (buffer A) and 0.1% formic acid in acetonitrile (buffer B) were utilized at a flow-rate of 0.8 mL/min. The following gradient was used: 0 min—5% buffer B; 16 min—45% buffer B; 21min—80% buffer B; 22 min—100% buffer B. The high-resolution SCIEX X500B QTOF electrospray ion source operated in negative ion mode.

SWATH analysis was carried out to provide a digital fingerprint of sample as previously described [21]. A total of 25 SWATH variable windows were utilized to obtain high-quality MS/MS spectra; the accumulation for the TOF MS was 0.150 s and the accumulation time for the TOF MS/MS was 0.025 s. The following MS source conditions were used: CUR = 40 psi, CAD = 11, IS = −4500 V, TEM = 450 °C, GS1 = 65 psi and GS2 = 60 psi.

Data were processed using SCIEX OS Software 2.1 (AB SCIEX GmbH, Landwehrstraße 54, Darmstadt, Germany). The SCIEX Natural Products 2.1 Library (AB SCIEX GmbH, Landwehrstraße 54, Darmstadt, Germany) was used for searching database compound spectra for matches to experimentally derived spectra.

### 2.3. Trolox^®^ Equivalents Antioxidant Capacity

Trolox^®^ equivalents antioxidant capacity of MOLE was evaluated spectrophotometrically, as previously described [30]. This assay evaluates the ability of cell lysates in preventing ABTS^+^ radical formation, compared to Trolox^®^ (vitamin E analogue) standards.

Briefly, 10 μL of cell lysates or Trolox^®^ standards (0.125–2.0 mM) were incubated in ABTS-met-Myo-PBS buffer and the absorbance at 734 nm was monitored for 2 min. The reaction was started by the addition of H_2_O_2_ (450 μM), followed for 10 min, and the variation of absorbance was then recorded. The variation of absorbance detected was compared to those obtained using Trolox^®^ standards. Cell lysate TAC were expressed as micromoles/mg protein tested.

An amount of 10 μL of MOLE stock solution dilutions (1/1000–1/500–1/100 and 1/10 working solution corresponding to 0.015, 0.075, 0.15 and 1.5 mg/mL of dried powder, respectively) were also tested and results obtained were comparable with those already reported [21] (data not shown).

### 2.4. Cell Cultures

C2C12 myoblasts (2 × 10^3^/cm^2^; ATCC, Manassas, VA, USA) were cultured in 25 cm^2^ culture flasks with Dulbecco’s modified Eagle’s medium (DMEM; HyClone, Oud-Beijerland, Holland) supplemented with Glutamax-I (4 mM l-alanyl-l-glutamine), 4.5 g/L glucose (Invitrogen, Carlsbad, CA, USA) and 10% heat-inactivated fetal bovine serum (FBS; HyClone). No antibiotics were used. The cells were incubated at 37 °C with 5% CO^2^ in a humidified atmosphere. Cells were split 1:6 twice weekly and fed 24 h before each experiment. Differentiation into myotubes was achieved by culturing preconfluent cells (85% confluency) in medium containing 2% FBS and monitoring them by microscopy and for myogenin and MHC expression by Western blot analysis [31].

Cells were treated with working solution 1/1000 or 1/100 MOLE or vehicle (methanol) in culture media for 24 h. Methylthiazolyldiphenyl-tetrazolium bromide (MTT) assay after MOLE treatment was performed and no statistically significant differences were found compared to untreated cells (data not shown).

At this working solution, the methanol concentration (0.1%, *v*/*v*) did not have any specific effect on myotubes. Each experiment was performed in triplicate. After each treatment, cells were trypsinized and centrifuged at 1200× *g* for 10 min at room temperature. Cells were then lysed and then utilized for biochemical analysis or tested for protein content using the Bradford method (Sigma-Aldrich, St. Louis, MO, USA).

### 2.5. Glutathione Homeostasis

Intracellular reduced (GSH) and oxidized (GSSG) glutathione contents were quantified by a DTNB–glutathione reductase recycling assay, as previously described [32]. 

Briefly, 10^7^ cells were collected and suspended in (1:1) (*v*/*v*) μL 5% sulfosalicylic acid (SSA). Cells were lysed by freezing and thawing three times and then were centrifuged at 10,000× *g* for 5 min at +4 °C. The deproteinized supernatant was then analyzed for total glutathione content. Oxidized glutathione was selectively measured in samples where reduced glutathione was masked by pretreatment with 2-vinylpyridine (2%). In total, 10 μL of sample was added to the reaction buffer (700 μL NADPH (0.3 mM), 100 μL DTNB (6 mM), 190 μL H_2_O). The reaction was started by adding 2.66 U/mL glutathione reductase and followed at 412 nm by the TNB stoichiometric formation. The variation of absorbance detected was compared to those obtained by using glutathione standards and results were normalized for protein content.

### 2.6. Preparation of Cell Homogenates and Western Blot Analysis

Following the indicated treatment, cellular extracts were immediately prepared as previously described [33,34]. The extracted proteins were used immediately or aliquoted and stored at −80 °C until used. Cellular proteins (10–20 μg in 25 mM Tris-HCl pH 8, 0.5% SDS, 0.05% 2-mercaptoethanol, 2.5% glycerol, and 0.001% bromophenol blue) were denatured at 100 °C for 5 min, subjected to SDS-PAGE in a 8–12% polyacrylamide gel, and then electroblotted onto a PVDF membrane at 130 V for 1 h. The blots were blocked with 5% non-fat dry milk (Bio-Rad Laboratories, Inc., Hercules, CA, USA) and then incubated with anti-Nrf2, anti-HO-1, anti-myogenin or anti-MHC antibodies (Santa Cruz Biotechnology, Santa Cruz, CA, USA). After washing and incubation with a horseradish peroxidase-conjugated secondary antibody, the blots were developed with ECL (Amersham Biosciences, GE Healthcare Europe GmbH, Glattbrugg, Switzerland). Bands were quantified using ImageJ software (Rasband, W.S., ImageJ, U.S. National Institutes of Health, Bethesda, MD, USA). The expression of β-actin (Sigma-Aldrich, St. Louis, MO, USA) was used to normalize the data.

### 2.7. Enzymatic Activities

Intracellular superoxide dismutase activity was measured using a commercial assay kit (Cayman Chemical Company, Ann Arbor, MI, USA) following the manufacturer’s instructions. Results were expressed as units/milligrams of protein tested [35].

Catalase activity was measured using a commercial assay kit (Cayman Chemical Company, Ann Arbor, MI, USA) following the manufacturer’s instructions. Results were expressed as units/milligrams of protein tested [36].

Glutathione peroxidase activity was measured using a commercial assay kit (Cayman Chemical Company, Ann Arbor, MI, USA) following the manufacturer’s instructions. Results were expressed as units/milligrams of protein tested [37].

Intracellular glutathione transferase activity was assayed with spectrophotometric methods as previously described [38]. In brief, cell extracts (100 μL) were incubated with 1 mM glutathione and 1 mM of 1-chloro-2,4-dinitrobenzene in 1 mL of 0.1 M potassium-phosphate buffer, pH 6.5. The enzymatic activity was monitored (37 °C) at 340 nm where the enzymatic product, the S-glutathionyl-2,4-dinitrobenzene, absorbed (ε340 nm = 9600 M/cm). For each spectrophotometric determination, the spontaneous reaction of glutathione with 1-chloro-2,4-dinitrobenzene was subtracted. GST activity was expressed in enzymatic units (U) at 37 °C and normalized for protein content.

### 2.8. Lipid and Protein Oxidation

*Thiobarbituric acid reactive substances.* TBARS levels were assayed by spectrophotometric analysis [35]. The methodology measured malondialdehyde (MDA) and other aldehydes produced by lipid peroxidation induced by hydroxyl-free radicals. Briefly, 150 µL cell lysate were added to 25 µL 0.2% butylated hydroxytoluene (BHT) and 600 µL 15% aqueous of trichloroacetic acid (TCA) in a 1.5 mL tube (Eppendorf, Hamburg, Germany). The mixture was centrifuged at 4000× *g* for 15 min at 4 °C. A total of 300 µL of the deproteinized supernatant was transferred in a Corning Cryotube 2 mL and added with 600 µL of TBA (0.375% in 0.25 M HCl). Samples were then heated at 100 °C for 15 min in boiling water. After cooling, sample absorbance was determined spectrophotometrically at 535 nm and compared to standard MDA (1,1,3,3-tetramethoxypropane) solution. The levels of TBARS were expressed in terms of nmol/mg protein.

*Protein carbonyls.* Protein carbonyl levels were determined by measuring the reactivity of carbonyl derivatives with 2,4-dinitrophenylhydrazine (DNPH) as described [39], with some modifications. In brief, cell lysates (100 μL) were precipitated with 10 volumes of HCl–acetone (3:100) (*v*/*v*), then washed with 5 mL HCl–acetone to remove chromophores. The protein pellet was then washed twice and disintegrated by hard vortexing during each wash and the supernatant was decanted after each centrifugation (800× *g*, for 20 min, 4 °C). Protein pellets were resuspended in 500 μL of PBS to which 500 μL of 10 mM DNPH (in 2 M HCl) were added and vortexed every 5 min for 30 min at room temperature. Protein blanks were prepared by adding 500 μL of 2 M HCl instead of DNPH. After mixing, 500 μL of 30% TCA were added to each tube, placed on ice for 10 min and then centrifuged (800× *g*, for 20 min, 4 °C). The supernatant was discarded and the pellets were washed with 20% TCA followed by three ethanol–ethylacetate (1:1) (*v*/*v*) washes in order to remove any unreacted DNPH. The pellets were then solubilized in 1 mL of 6 M guanidine hydrochloride and 20 mM potassium dihydrogen phosphate (pH 2.3). The carbonyl content was calculated from the absorbance measurement at 380 nm. Millimolar extinction coefficient ε380 = 22.00. Protein carbonyl content was expressed in terms of nmol/mg protein.

### 2.9. Statistical Analysis

The Kolmogorov–Smirnov test was used to evaluate the variable distribution. All data are expressed as means ± S.D. of three independent experiments, each performed in triplicate. A one-way ANOVA for repeated measures and Bonferroni post hoc analyses were used to determine significant variations among groups for each parameter evaluated. *p* < 0.05 was accepted as significant. The SPSS statistical package (Version 17.0 for Windows; SPSS Inc., Chicago, IL, USA) was used for statistical analysis. No statistical differences were found in all parameters analyzed between untreated controls and control vehicles (CTRLm) (data not shown).

## 3. Results

### 3.1. Metabolomic Fingerprint by UHPLC QTOF

Investigation of MOLE extract by UHPLC-MS lead to the detection of three main secondary metabolite groups: glucosinolates, flavonoids and phenolic acids. SWATH acquisition allowed to have a digital record of the sample and collect MS/MS spectral information for every detectable precursor in the defined mass range. Thus, the product ion spectral information was searched and matched against the SCIEX Natural Products Library 2.1 database for potential compound identification (library match score > 75% and mass error +/−2 ppm).

About 27 secondary metabolites among flavonoids, polyphenols and phenols were identified with a high library score (78–100%) [21].

A different approach was undertaken for glucosinolates (GLs) identification. These metabolites were missing of reference standards and MS/MS spectral information were present only for a few glucosinolates. GLs are thioglucoside compounds, containing a sulfated aldoxime moiety and a variable side chain derived from amino acids. This peculiar chemical structure leads typical fragment ions in the MS/MS spectra: in fact, diagnostic fragments at 259.01 *m*/*z* and 96.96 *m*/*z* can be assigned to the sulfated glucose moiety and the sulfate group, respectively [40,41].

Figure 1 reports extracted ion chromatograms XIC, MS and the SWATH MS/MS spectrum for identified glucosinolates: Glucosoonjnain (A; tR: 3.81), Glucomoringin (B; tR: 4.28), 4-*O*-acetylrhamnopyranosyloxybenzylGS (C; tR: 5.99, 6.4, 7.68) and 4-*O*-acetylglucopyranosyloxybenzylGS (D; tR: 6.98, 5.33). As shown, in every SWATH spectrum diagnostic fragment, ions were detected.

The metabolomic fingerprint highlighted the presence of glucosinolates, flavonoids and phenolic acids. The most intense peaks were represented by GLs. These metabolites are widely distributed in the Brassicaceae family and in the Moringa plant and are well known for their important biological activities, in particular the antioxidant and anti-inflammatory effects.

Table 1 shows the relative amounts of the components found in MOLE.

### 3.2. In Vitro Antioxidant Capacity of MOLE

Different dilutions of the methanolic extract of *Moringa oleifera* leaves stock solution were tested for antioxidant capacity. The Trolox^®^ equivalent antioxidant capacity of MOLE was determined. Results are shown in Figure 2. Dilutions of MOLE extract were linear up to 1/100 working solution. An amount of 1/1000 and 1/100 working solution were selected for cells treatment.

Cells were treated with MOLE 1/1000, MOLE 1/100 or vehicle for 24 h.

### 3.3. Evaluation of Redox Status after MOLE Treatment

No differences were found in the total antioxidant capacity after the C2C12 myotubes MOLE treatment as assessed by the modified Trolox^®^ equivalent antioxidant capacity assay (Figure 3).

Regarding glutathione homeostasis, compared to untreated cells, no differences were found in total intracellular glutathione (tGSH) after the 24 h MOLE treatment (Figure 3). Evaluating oxidized glutathione, a significant treatment effect was found (20% reduction for MOLE 1/100). No dose-dependent effect was found.

The resulting GSH/GSSG ratio was found increased in these samples if compared to untreated myotubes (25% and 29% increase for MOLE 1/1000 and 1/100, respectively) (*p* < 0.05, Figure 3).

### 3.4. Nrf2 and HO-1 Protein Expression

Compared to untreated cells, MOLE was able to increase Nrf2 and HO-1 protein levels in C2C12 myotubes. MOLE administration induced a treatment effect (*p* < 0.05) and a dose-dependent increase in Nrf2 (9% and 15% for MOLE 1/1000 and 1/100, respectively) and HO-1 (12% and 18% for MOLE 1/1000 and 1/100, respectively) protein levels (*p* < 0.05, Figure 4).

### 3.5. Evaluaion of Antioxidant Enzyme Activities after MOLE Treatment

Compared to untreated cells, MOLE was able to increase SOD, CAT, GPx and GST activity in C2C12 myotubes. MOLE administration induced a treatment effect (*p* < 0.05) and a dose-dependent increase in superoxide dismutase (8% and 24% for MOLE 1/1000 and 1/100, respectively) and glutathione transferase (11% and 17% for MOLE 1/1000 and 1/100, respectively) (*p* < 0.05, Figure 5). A treatment effect was found for catalase (17% increase for MOLE 1/100) and glutathione peroxidase (31% increase for MOLE 1/100) (*p* < 0.05, Figure 5). No dose effect was found for these enzymatic activities (Figure 5).

### 3.6. Evaluation of Oxidative Damage after MOLE Treatment

As shown in Figure 6, MOLE treatment did not induce any significant change in protein carbonyl and TBARS (malondialdehyde (MDA) and other aldehydes) levels if compared to untreated cells.

## 4. Discussion

In the present study, we found that the MOLE treatment increased the Nrf2 and HO-1 protein expression and induced antioxidant enzyme activities in the C2C12 skeletal muscle cell line. In particular, MOLE increased enzyme activities of catalase, superoxide dismutase, glutathione peroxidase and glutathione transferase. Regarding myotubes glutathione homeostasis, MOLE was able to increase the GSH/GSSG ratio, a main marker of redox status.

*Moringa oleifera* leaf extract may represent a nutritional supplement beneficial to skeletal muscle. As a whole, the components in *Moringa oleifera* leaf extract act as bifunctional molecule exerting antioxidant activity either directly by scavenging ROS or indirectly by inducing the antioxidant response and, therefore, ameliorating the redox status of the cells.

It must be considered that skeletal muscle is a tissue persistently exposed to a pro-oxidizing milieu due to its high rate of oxygen consumption and metabolic activity [42,43]. High levels of ROS and a decreased antioxidant defense lead to oxidative stress and, consequently, to an impairment of physiological functions. Sarcopenia, an age-related condition, as well as some pathological conditions are characterized by the loss of skeletal muscle mass, associated with decreased glutathione and increased oxidative stress [44,45,46].

To date, it has been clear that, while redox unbalance impairs muscle health and decreases muscular strength, the appropriate use of antioxidants would be beneficial to balance the ratio between oxidants and antioxidants in most of the physiopathological conditions [47,48,49].

We previously demonstrated that the increase in oxidative metabolism is mediated by the activation of the SIRT1/PPARα pathway eventually leading to mitochondrial biogenesis [21].

In recent years, SIRT1 has attracted extensive attention as a protein modulator of Nrf2, an important transcription factor that, through the binding to antioxidant-responsive elements (AREs), potentiates the cellular defense system and, therefore, prevents oxidative damage [50]. Nrf2 signaling also plays a key role in oxidative stress—mediates beneficial effects of exercise [51].

Among the various factors regulated by Nrf2, there is the heme oxygenase-1 (HO-1), an important enzyme catalyzing the rate-limiting step in heme degradation [52]. Carbon monoxide, a byproduct of HO-1 activity, is involved in mitochondrial oxidative phosphorylation and mitochondrial biogenesis [53,54]. Interestingly, we found that MOLE was able to increase the level of Nrf2 and HO-1 proteins in a dose-dependent manner.

Therefore, it is possible to speculate that one of the mechanisms by which MOLE enhances mitochondrial oxidative metabolism could also be supported by HO-1 induction through the Nrf2 pathway [55].

The mitochondrial electron transport chain represents the primary source of intracellular ROS, so an increase in oxidative metabolism naturally leads to an over-production of different forms of these reactive chemical molecules such as superoxide radical, hydrogen peroxide and hydroxyl radical [56].

It is noteworthy that HO-1 is an antioxidant enzyme and protects against oxidative stress, inflammation and metabolic dysregulation [57] and the upregulation of HO-1 expression is related to the reduction in ROS levels in various cells [58,59]. It has been shown that in human endothelial cells and in C2C12 myoblasts, the inhibition of HO-1 activity leads to an intracellular ROS increase [60]. Besides HO-1, other molecules are strongly involved in the antioxidant network responsible for maintaining the redox balance in the body.

The defense system consists of non-enzymatic antioxidants, such as glutathione (GSH), a potent reducing agent and a major antioxidant that maintains the cellular redox status, and enzymatic antioxidants, such as superoxide dismutase, catalase, glutathione peroxidase, glutathione transferase among others [56]. The antioxidant enzyme system is regulated by Nrf2; it modulates SOD that, in turn, catalyzes the dismutation of superoxide radicals to generate hydrogen peroxide; then, neutralizes in H_2_O through GPx in the cytosol. Hydrogen peroxide can also be eliminated by CAT in the mitochondria and peroxisomes. GST is involved in the detoxification of molecules through the formation of S-conjugates with GSH.

We speculate that MOLE in C2C12 myotubes, having the ability to increase the oxidative metabolism, exposes cells to an increased production of ROS and, interestingly, at the same time allows cells to counteract them through the upregulation of endogenous enzymatic activities. No oxidative damage was observed, in fact, the levels of TBARS and protein carbonyls remained unchanged.

An increased GST activity leads to a larger detoxification of molecules through their conjugation with GSH. Such an occurrence may result in a decrease in total glutathione levels; however, this phenomenon was not observed after MOLE treatment.

To explain these data, it should be considered that the optimal intracellular level of glutathione is as a result of synthesis, consumption, and regeneration, through an enzymatic network regulated by Nrf2.

The synthesis of GSH depends mainly on the activity of gamma-glutamylcysteine ligase (γ-GCL), which catalyzes the binding of glutamate to cysteine, the limiting step in the synthesis of GSH.

It must be pointed out that, Nrf2 controls both the expression of γ-GCL and the expression of the enzymes responsible for regenerating GSH levels, such as glutathione reductase (GR), which keeps the correct balance between reduced and oxidized glutathione.

We speculate that an increase in GR and γ-GCL in response to MOLE may confer to the cell the ability to regenerate, once it has been oxidized, and synthetize de novo the appropriate amount of GSH to maintain its level even though being used by GST increased activity.

A well-known marker of oxidative stress is represented by the ratio between the reduced and oxidized form of glutathione. Our data show that, an increase in the GSH/GSSG ratio was found after MOLE treatment, indicating that a better cellular redox state was reached despite the metabolic impulse induced by MOLE administration.

We also analyzed the myotubes redox status by evaluating the total non-enzymatic antioxidant capacity. In this assay, the combined antioxidant capacity between hydrophilic and lipophilic antioxidants was measured. Our results showed that intracellular TAC was unchanged after MOLE administration.

It should be noted that in the TAC assay, the contribution of important enzymes such as superoxide dismutase, glutathione peroxidase, and catalase [61,62], was not valued.

The identification of the bioactive molecules potentially responsible for the MOLE biological effect, through the metabolomic fingerprint (UHPLC QTOF analysis), highlighted the predominant presence of glucosinolates, flavonoids and phenolic acids [21]. These compounds activate the Nrf2-HO-1 pathway in different experimental models [63].

Previous reports using reporter gene assays revealed a significant increase in the ARE-dependent promoter activity of NAD(P)H dehydrogenase (quinone) 1 (NQO1) and GPx, indicating an activation of the Nrf2 pathway induced by MOLE glucosinolates. This effect was demonstrated both at the transcriptional and translational levels, in different cellular models (e.g., HepG2 cells, V79-MZ cells) and two Salmonella typhimurium strains [64]. Sulforaphane, derived by the glucosinolate glucoraphanin, and 3,3′-diindolylmethane derived by the glucosinolate glucobrassicin are antioxidant agents, effective in attenuating oxidative stress and tissue/cell damage in different in vivo and in vitro experimental models. They induce many cytoprotective proteins, including antioxidant enzymes such as HO-1, NAD(P)H: quinone oxidoreductase, glutathione transferase, gamma-glutamyl cysteine ligase, and glutathione reductase, through the Nrf2-antioxidant response element pathway [65,66].

Moreover, among biomolecules present in the MOLE extract, flavonoids and phenolics compounds, with their anti-inflammatory and antioxidant properties, represent good candidates to confirm the beneficial effects of MOLE on myotubes [21].

The flavonol quercetin promoted mitochondrial biogenesis in skeletal muscles and, therefore, ameliorated mitochondrial training adaptations and/or mitochondrial protein content, enzyme activities and/or respiratory function [67,68,69].

It was demonstrated that quercetin protects myotubes against TNFα-induced muscle atrophy under obese conditions through Nrf2-mediated HO-1 induction accompanied by the inactivation of NF-κB [52].

Differently from the above-mentioned works, we investigated the adaptive response to MOLE in the absence of any oxidant stimulus, showing for the first time a comprehensive framework on the ability of the extract, to modulate an antioxidant response in C2C12 myotubes.

Supplementation with antioxidants or plant extracts is a common practice among physically active people in order to limit the cellular damage potentially caused by enhanced metabolism/exercise-induced reactive species. However, these supplements are not always beneficial, and indeed, if not used in the correct way in terms of dose–time, they can interfere with the benefits of physical exercise [47]. Hence, considering the consumption of supplements by healthy people, it is especially interesting to understand the role of these molecules or plant extracts on the antioxidant network of skeletal muscle, a tissue particularly exposed to ROS, in order to evaluate any potential benefits or harmful effects [70,71,72].

## 5. Conclusions

In conclusion, our results provide evidence that MOLE, besides improving oxidative metabolism, has a beneficial effect on the antioxidant system of skeletal muscle cells, increasing the enzymatic capacity. This mechanism involved in C2C12 myotubes the induction of the Nrf2-HO-1 pathway.

Although our data are encouraging about the future use of MOLE for maintaining muscle health, further studies are recommended to better elucidate the action of this extract on the redox state under stressful conditions present in many physiological and pathological states.

## Figures and Tables

**Figure 1 molecules-26-05041-f001:**
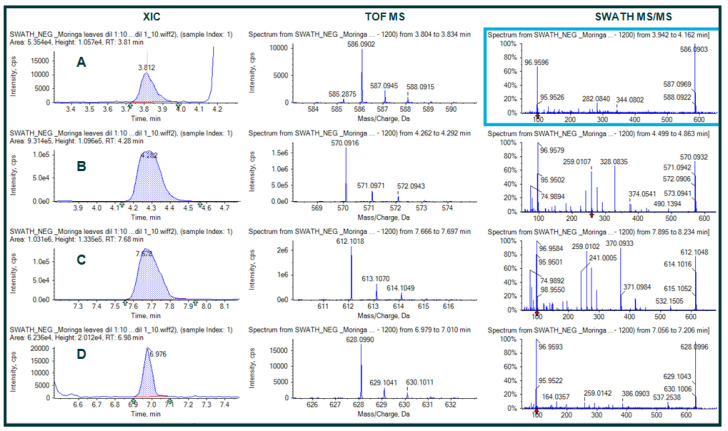
Extracted ion chromatograms XIC, MS and SWATH MS/MS spectrum of MOLE extract. (**A**) Glucosoonjnain (tR: 3.81), (**B**) Glucomoringin (tR: 4.28), (**C**) 4-*O*-acetylrhamnopyranosyloxybenzylGS (tR: 5.99, 6.4, 7.68) and (**D**) 4-*O*-acetylglucopyranosyloxybenzylGS (tR: 6.98, 5.33).

**Figure 2 molecules-26-05041-f002:**
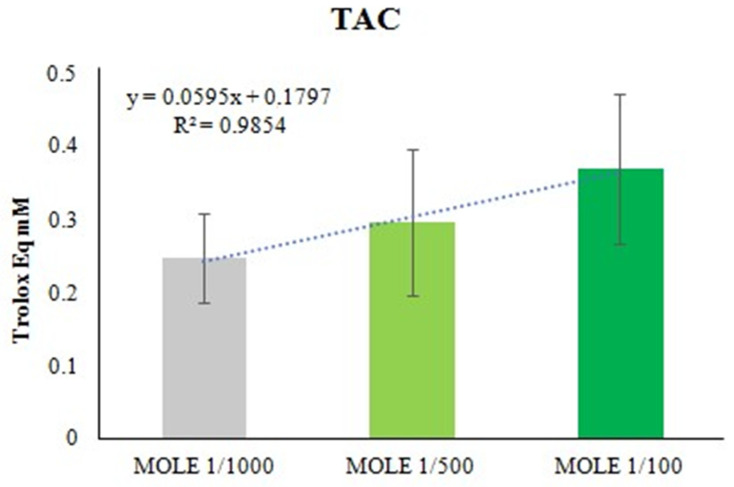
In vitro modified Trolox^®^-equivalent antioxidant capacity assay of MOLE. MOLE stock solution dilutions (1/1000–1/500 and 1/100 working solution) were tested and the variation of absorbance was recorded and compared to those obtained using Trolox^®^ standards (mM).

**Figure 3 molecules-26-05041-f003:**
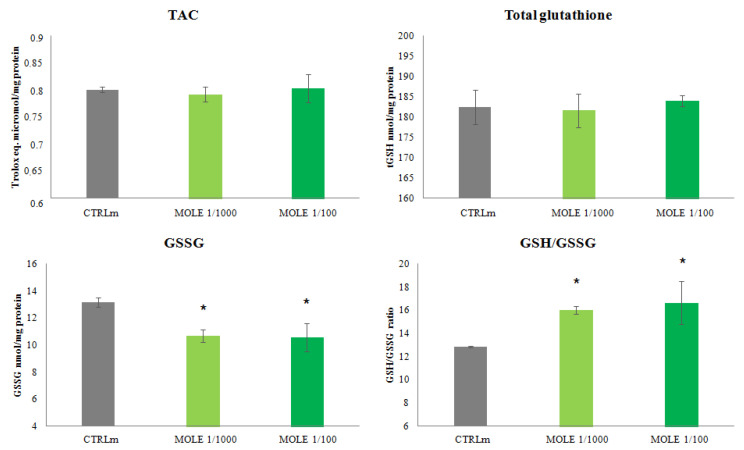
Total antioxidant capacity (TAC) and glutathione homeostasis analysis. Measurement of total antioxidant capacity (TAC), total glutathione (tGSH), oxidized glutathione (GSSG) and reduced to oxidized glutathione ratio (GSH/GSSG) was performed in C2C12 myotubes after 24 h MOLE (1/1000 and 1/100 working solutions) treatments. Data presented are the mean ± SD of three experiments. * *p* < 0.05 vs. CTRLm.

**Figure 4 molecules-26-05041-f004:**
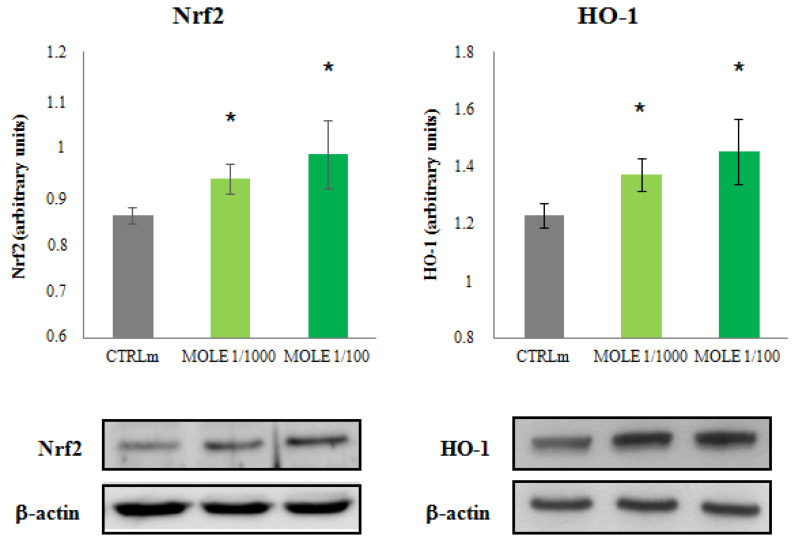
Western blot analysis. Measurement of Nrf2 and HO-1 protein levels analysis was performed in C2C12 myotubes after 24 h MOLE (1/1000 and 1/100 working solutions) treatments. Image J densitometric analysis of the Nrf2 and HO-1/β-actin ratios from the immunoblots is shown (the images are a representative of the experiments). Data presented are the mean ± SD of three experiments. * *p* < 0.05 vs. CTRLm.

**Figure 5 molecules-26-05041-f005:**
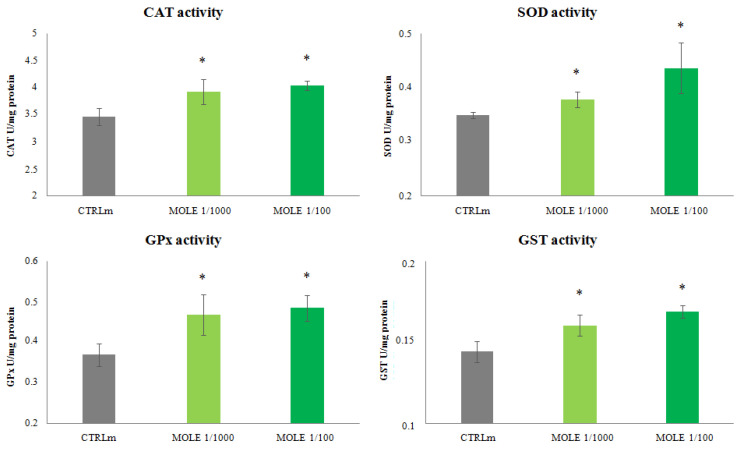
Catalase, superoxide dismutase, glutathione peroxidase and glutathione transferase activities analysis. Measurement of CAT, SOD, GPx and GST analysis was performed in C2C12 myotubes after 24 h MOLE (1/1000 and 1/100 working solutions) treatments. Data presented are the mean ± SD of three experiments. * *p* < 0.05 vs. CTRLm.

**Figure 6 molecules-26-05041-f006:**
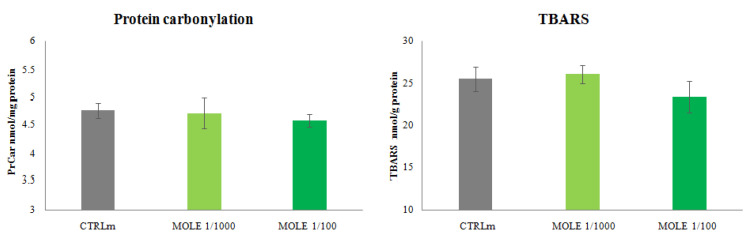
Thiobarbituric acid reactive substances (TBARS) and protein carbonyls (PrCar) analysis. Measurement of TBARS and PrCar analysis was performed in C2C12 myotubes after 24 h MOLE (1/1000 and 1/100 working solutions) treatments. Data presented are the mean ± SD of three experiments.

**Table 1 molecules-26-05041-t001:** Relative amounts of constituents in MOLE samples.

Components	Relative Amounts (%)
Glucomoringin	23.21
4-*O*-acetylrhamnopyranosyloxybenzylGS/7.67	22.16
Isoquercitrin	7.39
Rutin	5.28
Quercetin-*O*-β-d-glucose-acetate isomer/11.53	4.92
Neochlorogenic acid	4.57
Chlorogenic acid	4.57
Cryptochlorogenic acid	4.57
Astragalin/Luteoloside	4.22
4-*O*-acetylrhamnopyranosyloxybenzylGS/5.99	3.17
Quinic acid	2.81
Kaempferol-*O*-rutinoside	2.46
4-*O*-acetylrhamnopyranosylosxybenzylGS/6.41	2.11
Vitexin	2.11
Isovitexin	2.11
Quercetin-*O*-β-d-glucose-acetate isomer/11.98	1.06
Nepetin 7-glucoside	0.88
Quercetin-*O*-β-d-glucose-acetate isomer/12.34	0.70
Orientin	0.53
4-*O*-acetylglucopyranosyloxybenzylGS/6.97	0.18
Esculin	0.18
Isorhamnetin-*O*-neohespeidoside	0.18
Glucosoonjnain	0.18
Quercetin	0.04
Citric acid	0.04
Sinalbin	0.04
Kaempferol	0.04
Kaempferol 3-*O*-(3″,4″-di-*O*-acetyl-α-l-rhamnopyranoside)	0.04
Isorhamnetin	0.04
Vitamin B2	0.04
Quercetin-di-*O*-glucoside	0.04
Pueranin	0.04
4-*O*-acetylglucopyranosyloxybenzylGS/5.34	0.04
4-*O*-acetylglucopyranosyloxybenzylGS/6.16	0.04
Ferulic/Isoferulic acid	0.04
Protocatechuic acid	0.04

## Data Availability

Not applicable.

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
