# Peer review of "Moringa oleifera Leaf Extract Upregulates Nrf2/HO-1 Expression and Ameliorates Redox Status in C2C12 Skeletal Muscle Cells"

_molecules, 2021, doi:10.3390/molecules26165041_

Round 1

Reviewer 1 Report

This manuscript (Moringa Oleifera leaf extract upregulates Nrf2/HO-1 expression and ameliorates redox status in C2C12 skeletal muscle cells.) is interesting observation that it shows the antioxidant properties of Moringa oleifra Lam (MOLE) on skeletal muscle cells. I think however that there are a few improvements that should be made before publication.

Comment

1) MOLE increases the enzymatic activity of several antioxidant proteins, such as CAT, SOD, and GST in C2C12 cells (Fig. 5). However, MOLE has no influence on TBARS levels, a measure of oxidative stress (Fig. 6). If MOLE increases these antioxidant activity, then I think that MOLE would decreases TBARS levels.

2) There is a mixture of MOLE and MOR in Figure. The authors should unify to MOLE.

3) Oxidative stress inhibits muscle differentiation of C2C12 myoblast. The authors should show the effect of MOLE on muscle differentiation of C2C12 in this manuscript.

Author Response

Reviewer 1

This manuscript (Moringa Oleifera leaf extract upregulates Nrf2/HO-1 expression and ameliorates redox status in C2C12 skeletal muscle cells.) is interesting observation that it shows the antioxidant properties of Moringa oleifra Lam (MOLE) on skeletal muscle cells. I think however that there are a few improvements that should be made before publication.

We thank the reviewer for his comments that help us improve the quality of our manuscript.

As request, the manuscript was checked by a native English speaker.

Replies to comments are listed below.

Comment

1) MOLE increases the enzymatic activity of several antioxidant proteins, such as CAT, SOD, and GST in C2C12 cells (Fig. 5). However, MOLE has no influence on TBARS levels, a measure of oxidative stress (Fig. 6). If MOLE increases these antioxidant activity, then I think that MOLE would decreases TBARS levels.

We thank the reviewer for the comment.

Our data clearly indicate that supplementation with MOLE is able to increase the activity of enzymatic antioxidant systems through the induction of Nrf2/HO-1 axis. In the discussion we stated: “We speculate that MOLE in C2C12 myotubes, having the ability to increase oxidative metabolism, exposes cells to an increased production of ROS and interestingly, at the same time allows cells to counteract them through the upregulation of endogenous enzymatic activities. No oxidative damage was observed, in fact, the levels of TBARs and protein carbonyls remained unchanged.

In fact, in our experimental model, cells are not in oxidative stress condition, hence oxidative marker levels do not change. Our results allow us to speculate that MOLE supplementation, ameliorating the antioxidant defense system and redox status, makes cells capable to recover more efficiently when submitted to an eventual redox imbalance.

2) There is a mixture of MOLE and MOR in Figure. The authors should unify to MOLE.

We thank the reviewer for the comment.

We have now standardized the nomenclature in the figures.

3) Oxidative stress inhibits muscle differentiation of C2C12 myoblast. The authors should show the effect of MOLE on muscle differentiation of C2C12 in this manuscript.

We thank the reviewer for the comment. This is an interesting point.

Our hypothesis was that MOLE could allow skeletal muscle cells to improve their redox status counteracting efficiently ROS induced by oxidative metabolism, stimulated by MOLE itselfs (as we reported in a previous paper [doi: 10.1016/j.phyplu.2020.100014.],  through the antioxidant enzymatic system induction.

Indeed, WE demonstrated in this paper that MOLE exerts antioxidant activity both directly by scavenging ROS and indirectly by inducing the antioxidant response and therefore ameliorating the redox status of the cells.

A study on C2C12 differentiation by MOLE was beyond the scope of this work and indeed your comment provides us with a starting point for future perspectives.

Reviewer 2 Report

Manuscript entitled "Moringa oleifera leaf extract upregulates Nrf2/HO-1 expression and improves redox status in C2C12 skeletal muscle cells" suggested the antioxidant activity of MOLE in skeletal muscle cells and provided a possible mechanism such as Nrf2/HO-1 upregulation.

Interestingly, authors conducted in vitro assay on skeletal muscle cells (C2C12), therefore the results presented on the paper are expected to be applicable to sarcopenia, which is being actively studied in recent years.

However, a detailed description of the background of the discovery of Nrf2/HO-1 among various transcription factors is insufficient.  Therefore, the authors should mention the expression levels of other transcription factors involved in ROS mechanisms or SIRT, if it had measured.

Minor comments:

  1. Period on the title needs to be deleted
  2. Line 244: GL needs to be presented on the abbreviation
  3. It is better to present MSMS to MS/MS (Line 243, 246, 249, and 256)
  4. Line 197: 0,2% should be 0.2%
  5. Line 198: 1,5 ml should be 1.5 ml

Author Response

Reviewer 2

Manuscript entitled "Moringa oleifera leaf extract upregulates Nrf2/HO-1 expression and improves redox status in C2C12 skeletal muscle cells" suggested the antioxidant activity of MOLE in skeletal muscle cells and provided a possible mechanism such as Nrf2/HO-1 upregulation.

We thank the reviewer for his comments that help us improve the quality of our manuscript.

Replies to comments are listed below.

Interestingly, authors conducted in vitro assay on skeletal muscle cells (C2C12), therefore the results presented on the paper are expected to be applicable to sarcopenia, which is being actively studied in recent years.

However, a detailed description of the background of the discovery of Nrf2/HO-1 among various transcription factors is insufficient.  Therefore, the authors should mention the expression levels of other transcription factors involved in ROS mechanisms or SIRT, if it had measured.

We thank the reviewer for the comment.

In Discussion section, we stated that we have previously demonstrated that MOLE is able to increase C2C12 myotubes oxidative metabolism by the activation of SIRT1/PPAR? pathway [doi: 10.1016/j.phyplu.2020.100014.].

SIRT1 is a modulator able to counteract oxidative stress via the activation of Nrf2, an important transcription factor that, through the binding to antioxidant-responsive elements (AREs), potentiates cellular defense system and therefore preventing oxidative damage [doi: 10.3390/cancers12071822.].  

The aim of the work was to demonstrate the ability of MOLE to modulate the C2C12 myotubes redox status and to improve cellular enzymatic antioxidant defenses. Our data indicate that supplementation with MOLE increases the activity of enzymatic antioxidant systems and through the induction of SIRT1-Nrf2-HO-1 axis. Furthermore, we have demonstrated an improvement of the redox status in C2C12 myotubes.

Minor comments:

Period on the title needs to be deleted

Line 244: GL needs to be presented on the abbreviation

It is better to present MSMS to MS/MS (Line 243, 246, 249, and 256)

Line 197: 0,2% should be 0.2%

Line 198: 1,5 ml should be 1.5 ml

We thank the reviewer for pointing out these inaccuracies.

Now we have edited the text according to his comments.